# An evaluation of RNA-seq differential analysis methods

**Dongmei Li**[1]*, **Martin S. Zand**[1,2], **Timothy D. Dye**[3], **Maciej L. Goniewicz**[4], **Irfan Rahman**[5], **Zidian Xie**[1]

1 Clinical and Translational Science Institute, School of Medicine and Dentistry, University of Rochester, Rochester, NY, United States of America, 2 Department of Medicine, Division of Nephrology, School of Medicine and Dentistry, University of Rochester, Rochester, NY, United States of America, 3 Department of Obstetrics and Gynecology, School of Medicine and Dentistry, University of Rochester, Rochester, NY, United States of America, 4 Department of Health Behavior, Roswell Park Comprehensive Cancer Center, Buffalo, NY, United States of America, 5 Department of Environmental Medicine, School of Medicine and Dentistry, University of Rochester, Rochester, NY, United States of America

* Dongmei_Li@urmc.rochester.edu

**Data Availability Statement:** The summarized RNA-seq count data from two mouse strains in neuroscience research can be downloaded from http://bowtie-bio.sourceforge.net/recount/countTables/bottomly_count_table.txt. The R code

## Abstract

RNA-seq is a high-throughput sequencing technology widely used for gene transcript discovery and quantification under different biological or biomedical conditions. A fundamental research question in most RNA-seq experiments is the identification of differentially expressed genes among experimental conditions or sample groups. Numerous statistical methods for RNA-seq differential analysis have been proposed since the emergence of the RNA-seq assay. To evaluate popular differential analysis methods used in the open source R and Bioconductor packages, we conducted multiple simulation studies to compare the performance of eight RNA-seq differential analysis methods used in RNA-seq data analysis (edgeR, DESeq, DESeq2, baySeq, EBSeq, NOISeq, SAMSeq, Voom). The comparisons were across different scenarios with either equal or unequal library sizes, different distribution assumptions and sample sizes. We measured performance using false discovery rate (FDR) control, power, and stability. No significant differences were observed for FDR control, power, or stability across methods, whether with equal or unequal library sizes. For RNA-seq count data with negative binomial distribution, when sample size is 3 in each group, EBSeq performed better than the other methods as indicated by FDR control, power, and stability. When sample sizes increase to 6 or 12 in each group, DESeq2 performed slightly better than other methods. All methods have improved performance when sample size increases to 12 in each group except DESeq. For RNA-seq count data with log-normal distribution, both DESeq and DESeq2 methods performed better than other methods in terms of FDR control, power, and stability across all sample sizes. Real RNA-seq experimental data were also used to compare the total number of discoveries and stability of discoveries for each method. For RNA-seq data analysis, the EBSeq method is recommended for studies with sample size as small as 3 in each group, and the DESeq2 method is recommended for sample size of 6 or higher in each group when the data follow the negative binomial distribution. Both DESeq and DESeq2 methods are recommended when the data follow the log-normal distribution.

for simulations and real data analysis could be found from the GitHub repository https://github.com/DongmeiLi2017/RNA-seq-Analysis-Methods-Comparison.

**Funding:** This work is supported by the University of Rochester's Clinical and Translational Science Award (CTSA) number UL1 TR000042, UL1 TR002001, and U24TR002260 from the National Center for Advancing Translational Sciences of the National Institutes of Health (Drs. Li and Zand). Dr. Zand is also supported by the National Institute of Allergy and Infectious Diseases and the National Institute of Immunology, grant numbers AI098112 and AI069351. This study was supported by the National Institute of Environmental Health Sciences with grant number NIH 1R21ES032159-01A1 and National Institute on Aging with grant number NIH 1U54AG075931-01 (Drs. Li and Rahman). This study was supported by the grants from the WNY Center for Research on Flavored Tobacco Products (CRoFT) under cooperative agreement U54CA228110 which is supported by the National Cancer Institute of the National Institutes of Health (NIH) and the Food and Drug Administration (FDA) Center for Tobacco Products (Drs. Li, Goniewicz, Rahman, Xie). The content is solely the responsibility of the authors and does not necessarily represent the official views of the NIH and FDA. The funders had no role in study design, data collection and analysis, decision to publish, or preparation of the manuscript.

**Competing interests:** The authors have declared that no competing interests exist.

# Introduction

High-throughput transcriptome sequencing technologies have profound impact on our ability to address an increasingly diverse range of biological and biomedical problems, and improve our understanding of human diseases by capturing an accurate picture of molecular processes within the cell [1]. RNA-seq has become a major assay for measuring relative transcript abundance and diversity, and has been used as a standard tool for the life sciences research community [2]. In RNA-seq experiments, RNAs are extracted from cells, complementary DNA (cDNA) is made from the RNA sample and sequenced, producing millions of reads. The reads are then mapped to the reference genome, and the total count of reads to a gene is used as a measure for this gene's expression level. The analysis of RNA-seq data has its own challenges. For example, the read coverage may not be distributed uniformly along the genome due to the variation in nucleotide composition in different genomic regions. In addition, longer genes have more mapped reads than shorter genes with the same expression level, which is usually ignored in differential analysis by assuming the effect of gene length is the same across all samples. However, this difference may affect the ranking of significant genes when selecting candidate genes for further verification.

Before differential expression analysis, the summarized feature counts need to be pre-processed which includes trimming, filtering, and normalizing. Current methods for RNA-seq gene differential expression analysis include both parametric methods and nonparametric methods. Parametric methods implemented in open source packages include edgeR [3], DESeq [4], DESeq2 [5], NBPSeq [6], TSPM [7], baySeq [8], EBSeq [9], ShrinkSeq [10], Voom/vst [11, 12] in R or Bioconductor, and Cuffdiff2 in Cufflinks [13]. Nonparametric methods include SAMseq [14] and NOIseq [15] in R or Bioconductor. Several comparative studies have been conducted to compare the performance of different RNA-Seq analysis methods through simulation studies and using real RNA-Seq data [16–18]. Soneson and Delorenzi [16] compared eleven RNA-Seq differential analysis methods for their true positive rate and false discovery rate using simulated data generated from negative binomial distributions. The eleven RNA-Seq differential analysis methods include DESeq, edgeR, NBPSeq, TSPM, baySeq, EBSeq, NOISeq, SAMSeq, ShrinkSeq, voom(+limma), and vst(+limma). The comparison results indicated some problems for all eleven methods when the sample size is very small. When sample size is large, they recommended the Voom/vst method and the SAMseq method that performed relatively well under many different conditions. Seyednasrollah et al. [18] compared eight software packages for RNA-Seq differential analysis including Cuffdiff 2, which was not included in previous comparisons. The eight software packages including edgeR, DESeq, baySeq, NOIseq, SAMseq, limma, and Cuffdiff2 were compared using two publicly available real RNA-Seq datasets for total number of rejections and estimated proportion of false discoveries. The comparison of eight RNA-Seq diffrential analysis methods detected the large variation between the methods and recommended limma and DESeq as the safest choice when sample size is very small (below 5 in each group). DESeq2 [5] is a successor to the DESeq method with flexibility to accommodate more complexed study design of sequencing experiments. As more statistical methods for gene differential analyses have been developed in recent years, there is a need for an update on the differential analysis methods comparisons.

Here, we present a systematic evaluation of the performance of eight popular RNA-seq differential analysis methods including edgeR, DESeq, DESeq2, baySeq, EBSeq, Voom, SAMSeq and NOISeq, implemented in R or Bioconductor, as indicated by false discovery rate (FDR) control, power, and stability. Our comparisons are conducted on both simulated data and publicly available real RNA-seq data. To ensure all compared methods are invariant to library size, we evaluate performance under two library size scenarios: equal library size across all samples;

and unequal library size across samples. We illustrated the large differences between different RNA-seq analysis methods applied to the same data set and provided empirical evidence and advice to investigators regarding selection of RNA-seq data analysis methods, taking into account the distribution assumption, sample size and acceptable FDR.

## Materials and methods

For our purposes, consider the null hypothesis of no differential gene expression. Among $m$ hypothesis tests, $m_0$ represent cases where no differential expression exists; i.e. these are "true null hypotheses". $R$ represents the number of rejected null hypotheses, and $V$ represents the number of tests that result in false rejections (i.e., $V$ represents the number of false discoveries). $S$ represents the number of tests that result in true rejections (i.e. the number of true discoveries). Following Benjamini and Hochberg [19] we define the false discovery rate as:

$$FDR = E(\frac{V}{R}|R > 0)Pr(R > 0). \tag{1}$$

Power and stability are defined as follows [20]:

$$Power = E(\frac{S}{m - m_0}|m > m_0), \tag{2}$$

$$Stability = SD(R). \tag{3}$$

Power is defined as the expected proportion of identified differentially expressed genes among all the truly differentially expressed genes, given at least one genes are truly differentially expressed in the data. Stability is defined as the standard deviation of total rejections. Power is defined as 0 and Stability becomes a measure of standard deviation (SD) of false detections when $m = m_0$.

### edgeR

edgeR [3] assumes the summarized count data $Y_{gi}$ for $g$th gene $i$th sample follow a negative binomial distribution as follows.

$$Y_{gi} \sim NB(M_i p_{gj}, \phi_g), \tag{4}$$

where $M_i$ is the library size for $i$th sample, $p_{gj}$ is the relative abundance for $g$th gene and $j$th group, and $\phi_g$ is the dispersion for $g$th gene. The mean and variance of the summarized count for $g$th gene are

$$E(Y_{gi}) = \mu_{gi} = M_i p_{gj}, \quad and \tag{5}$$

$$Var(Y_{gi}) = \sigma_{gi}^2 = \mu_{gi}(1 + \mu_{gi}\phi_g). \tag{6}$$

For differential analysis between two groups, we test $H_0$: $p_{g1} = p_{g2}$ for each gene. edgeR uses an empirical Bayes procedure moderating the degree of overdispersion across genes. Conditioning on the total count for the gene, the conditional maximum likelihood method is used to estimate genewise dispersion. Then, the dispersions are shrunk towards a common value through borrowing information between genes using the empirical Bayes procedure. For each gene, the differential expression between groups is assessed using an exact test analogous to Fisher's exact test taking the overdispersion into account. edgeR can also fit a negative

binomial generalized log-linear model to the read counts for each gene, and conduct genewise statistical tests using likelihood ratio tests.

## DESeq and DESeq2

DESeq [4] extended the negative binomial model proposed in edgeR and linked the variance and mean using a more general and data-driven relationship as follows.

$$\sigma_{gi}^2 = \mu_{gi} + M_i^2 v_{gj}, \tag{7}$$

where $\mu_{gi}$ is the random error or "shot noise" term and $M_i^2 v_{gj}$ is a raw variance term. $v_{gj}$ is a smooth function of $p_{gj}$ with $v_{gj} = v(p_{gj})$, which models the dependence of the raw variance on the relative abundance of the $g$th gene. DESeq conducts the differential expression test $H_0$: $p_{g1} = p_{g2}$ for each gene by an exact test analogous to Fisher's exact test, in which the test statistics are defined as the total count in each group, and the sum of those total counts across groups. The $p$ value is calculated as the sum of probabilities of observing an as extreme as or more extreme value of the total count in the treatment group, given that the total count across groups is fixed.

DESeq2 [5], a successor to the DESeq method, uses a generalized linear model (GLM) approach to accommodate more complex designs to model the relationship between the relative abundance and group difference. The approach uses a logarithm link between relative gene abundance and a design matrix as follows:

$$log p_{gj} = \sum_r x_{jr} \beta_{gr}. \tag{8}$$

In DESeq2, the dispersion $\phi_g$ is assumed to follow a log normal prior distribution $\phi_g \sim N(log \phi_{trend}(\bar{\mu}_g), \sigma_d^2)$. $\phi_{trend}(\bar{\mu}_g)$ is a function of the mean of normalized count for the $g$th gene, and $\sigma_d^2$ is the true dispersion around the trend that is common for all genes. DESeq2 integrates the dispersion estimate and fold change estimate from an empirical Bayes approach and tests differential expression using a Wald test.

## baySeq

baySeq [8] detects differentially expressed genes by estimating the posterior probability of a model using the observed data and an empirical Bayes approach. The baySeq method assumes data follow a negative binomial distribution and use an empirically determined prior distribution derived from the whole dataset. The maximum likelihood method is used to estimate data dispersion. A posterior probability of non-differential expression and a Bayesian FDR estimate are produced by the baySeq method to select significantly differentially expressed genes.

## EBSeq

EBSeq [9] was developed for identifying differentially expressed isoforms, but has been shown to be a robust approach for identifying differentially expressed genes. EBSeq assumes that within condition $C$, the expected count for the $g$th gene, $i$th sample $Y_{gi}^C$ follows a negative binomial (NB) distribution:

$$Y_{gi}^C | r_{g,0} l_i, q_g^C \sim NB(r_{g,0} l_i, q_g^C), \tag{9}$$

where $l_i$ denotes the library size in the $i$th sample and $r_{g,0}$ denotes the baseline expression level. The mean expression level for the $g$th gene within condition $C$ is $u_g^C$, which is equal to

$r_{g,0}(1 - q_g^C)/q_g^C$, and the variance for $g$th gene within condition $C$ is $(\sigma_g^C)^2$, which is equal to $r_{g,0}(1 - q_g^C)/(q_g^C)^2$. The prior distribution of $q_g^C$ is $q_g^C \sim Beta(\alpha, \beta)$, where $\alpha$ and $\beta$ are obtained through the expectation-maximization (EM) algorithm. For a two-group comparison, EBSeq tests $H_{0g} : \mu_g^{C_1} = \mu_g^{C_2}$ based on the negative binomial-beta empirical Bayes model and obtains the posterior probability of being differentially expressed via Bayes' rule using the EM algorithm. EBSeq also provides a Bayesian FDR estimate to facilitate the selection of significantly differentially expressed genes.

## Voom

Unlike the negative binomial model approach, the voom method takes a linear modeling strategy to model count data [12]. Voom defines the log-counts per million (log-cpm) value $y_{gi}^*$ as:

$$y_{gi}^* = log_2(\frac{y_{gi} + 0.5}{M_i + 1.0} \times 10^6), \tag{10}$$

where $M_i = \sum_{g=1}^{G} Y_{gi}$. Assume $var(y_{gi}) = \mu_{gi} + \phi\mu_{gi}^2$. Then, conditional on $M_i$, $var(y_{gi}^*) = var(log_2(y_{gi}))$ since $M_i$ was treated as a constant. Based on the delta rule and Taylor's theorem, $var(y_{gi}^*) = var(\mu_{gi} + \frac{y_{gi} - \mu_{gi}}{\mu_{gi}}) = \frac{var(y_{gi})}{\mu_{gi}^2} = \frac{1}{\mu_{gi} + \phi}$, where $\mu_{gi} = E(y_{gi})$. $\mu_{gi}$ is estimated by the following equation by fitting a LOWESS curve.

$$\hat{\mu}_{gi} = \hat{\mu}_{gi}^* + log_2(M_i + 1.0) - log_2(10^6), \tag{11}$$

where $\hat{\mu}_{gi}^* = E(y_{gi}^*)$. The piecewise linear function $lo(\hat{\mu}_{gi})$ defined by the LOWESS curve is the predicted square-root standard deviation of $y_{gi}^*$. The voom precision weight is the inverse variances $w_{gi} = lo(\hat{\mu}_{gi})^{-4}$. The log-cpm values $y_{gi}^*$ and associated weights $w_{gi}$ are used as input in the limma empirical Bayes analysis pipeline. Moderated $t$-statistics are used for gene differential expression analysis.

## SAMseq

SAMseq [14] is a nonparametric method proposed for RNA-seq count data differential analysis that is not based on Poisson or negative binomial models. SAMseq uses the two-sample Wilcoxon rank statistic as follows for two-group comparisons:

$$T_g^* = \frac{1}{B}\sum_{b=1}^{B}\sum_{t \in C_1}R_{gt}(Y'^b) - \frac{n_1(n+1)}{2}, \tag{12}$$

where $R_{gt}(Y'^b)$ is the rank of $Y_{gi}'$ in resampling $b$ in the first group $C_1 = \{i$: Sample $i$ is from group 1$\}$. $Y_{gi}' \sim Poisson(\frac{(\prod_{i=1}^{n} M_i)^{1/n}}{M_i} Y_{gi})$ are resampled counts from the Poisson distribution. $n_1$ is the sample size for group 1 and $n = n_1 + n_2$ is the total sample size of the two groups. Simulation studies have shown that $B = 20$ is large enough to give stable value of $T_g^*$ with sufficient power. A larger absolute value of $T_g^*$ indicates stronger evidence of differential expression for the $g$th gene between groups. The resampling from the Poisson distribution is used to account for different sequencing depths. The permutation method is then used to generate the null distribution of the Wilcoxon rank statistic and to estimate the FDR.

## NOIseq

NOIseq [15] takes sequencing-depth corrected and normalized RNA-seq count data and models the noise distribution by contrasting the logarithm of fold change and absolute expression differences between groups. NOIseq considers a gene to be differentially expressed if the corresponding logarithm of fold change $Fold_g = log_2(Y_g^{c_1}/Y_g^{c_2})$ and absolute expression differences $D_g = |Y_g^{c_1} - Y_g^{c_2}|$ have a high probability of being higher than noise values, where the threshold probability is 0.8. The probability distribution of $Fold_g$ and $D_g$ in noise data is computed by contrasting gene counts within the same condition $c_1$ or $c_2$.

## Data extraction

The RNA-seq count data from real data examples are downloaded from the ReCount website [21]. One example is to identify differentially expressed genes between two commonly used inbred mouse strains in neuroscience research: C57BL/6J(B6) and DBA/2J(D2) [22]. Summarized count data from the RNA-seq experiment are from 10 B6 mouse samples and 11 D2 mouse samples. The R code for simulations and real data analysis could be found from the Github repository https://github.com/DongmeiLi2017/RNA-seq-Analysis-Methods-Comparison.

## Results

### Simulation study

**Negative binomial distribution.** We conducted simulation studies to compare FDR control, power and stability of eight commonly used RNA-seq differential analysis methods: edgeR, DESeq, DESeq2, baySeq, EBSeq, Voom, SAMSeq and NOISeq. The edgeR Exact approach using the exact method and the edgeR GLM approach using general linear models are both included. We considered two different library size scenarios to examine the effect of sequencing depth on gene differential analysis. In one scenario, all samples have equal library size of $11 \times 10^6$; in the other scenario samples have either library size $2 \times 10^7$ or $1 \times 10^7$ alternatively. Summarized RNA-seq count data are simulated from a negative binomial distribution with mean $\mu_{gk}$ and variance $\mu_{gk} + \mu_{gk}^2/\theta_g$ for $g$ the gene $k$th sample using the *rnegbin* function in R. Here, $\mu_{gk} = E(Y_{gk}) = M_k p_{gk}$, where $Y_{gk}$ is the summarized count data for $g$ the gene $k$th sample, $M_k$ is the library size for $k$th sample, $p_{gk}$ is the relative abundance for $g$th gene $k$th sample, and $\theta_g$ is the dispersion for $g$th gene. Both $p_{gk}$ and $\theta_g$ are estimated from real RNA-seq data on two non-cancerous neural stem cells from two different subjects [23]. The RNA-seq count data from the neural stem cells are downloaded from the supplemental data files of the DESeq methods paper [4]. For each simulation study, we included 100 independently generated two-group comparison samples with sample size ($n$) of 3, 6, and 12 in both the cancer and normal groups. For all simulation studies, we set the total number of genes ($m$) as 10,000. The fractions of truly differentially expressed genes ($\pi_1 = \frac{m-m_0}{m}$) were set at 1%, 5%, 10%, 25%, 50%, 75%, or 90% to cover different scenarios. Among the genes that are assumed to be differentially expressed in our simulation studies, half of genes in the cancer group are up-regulated with a doubled expression level, and another half of genes are down-regulated with 50% expression levels, compared to the normal group. The simulated RNA-seq count data are first filtered by examining the total count from all samples for each gene. Genes with total count less than 10 are filtered out from differential analysis. All remaining genes are then normalized by the trimmed mean of M values (TMM) normalization method [24].

**Log-normal distribution.** The RNA-seq count data simulated from the negative binomial distribution could be biased in favor of differential analysis methods based on negative

binomial distribution assumption, thus a second simulation study is conducted to evaluate the performance of eight RNA-seq differential analysis methods using simulated RNA-seq count data generated from log-normal distribution. In this simulation study, the library sizes are unequal. Due to minor differences resulting from simulations based on negative binomial distribution, we did not conduct simulations with equal library sizes. Summarized RNA-seq count data are generated from log-normal distribution using the *rlnorm* function in R with mean $\mu_{gj}$ and standard deviation $\sigma_g$ for $g$th gene and $j$th group ($g = 1, \ldots, m$ and $j = 1, 2$ with 1 for cancer group and 2 for normal group). Moreover, $\mu_{gj} = fc_j \cdot log(\frac{E(Y_{gk})+0.5}{\sum E(Y_{gk})+1.0} \cdot 10^6)$. The *fc* denotes the fold change factor: $fc_1 = 2$ for up-regulated genes, $fc_1 = 0.5$ for down-regulated genes, and $fc_1 = 1$ for unchanged genes in the cancer group; $fc_2 = 1$ for all genes in the normal group. In $\mu_{gj}$, $E(Y_{gk})$ is the mean of summarized count data from two non-cancerous neural stem cells from two different subjects [4] for $g$th gene in $k$th sample. The standard deviation $\sigma_g$ is computed through $\sigma_g = \frac{\sigma_{Y^*_{gk}} + 1/E(Y_{gk})}{10}$, where $Y^*_{gk} = log(\frac{Y_{gk}+0.5}{\sum Y_{gk}+1.0} \cdot 10^6)$ with $Y_{gk}$ denoting the summarized count data from the neural stem cells in the normal group for $g$th gene in $k$th sample. The total number of independently generated two-group comparison samples is 20 for each of two group comparisons with sample size ($n$) of 3, 6, and 12 in each group. In each simulation, there are 1000 genes with proportion of truly differentially expressed genes ($\pi_1 = \frac{m-m_0}{m}$) equalling to 5%, 10%, 15%, or 20% to mimic the situation in reality. Same filtering criteria and normalization algorithms are used for the simulation with log-normal distribution as that for the negative binomial distribution.

## Simulation results

**Simulation results from negative binomial distribution.** Simulation results for FDR, power, total discoveries, and SD of total discoveries from eight RNA-seq differential analysis methods with equal and unequal library sizes for all samples are shown in Figs 1 and 2. When sample size is equal to 3 in each group, all methods have large estimated FDR. Further, FDR estimates vary widely across different proportions of truly differentially expressed genes, ranging from 1% to 90% (S1 Table). The median estimated FDR from the DESeq2, baySeq, EBSeq, SAMSeq, and NOISeq methods is relatively smaller than the corresponding median from the edgeR, DESeq, and Voom methods. Power comparisons show that the DESeq2, EBSeq, SAMSeq, and NOISeq have relatively larger power than other methods (S2 Table). Among four methods with relatively larger power, the EBSeq method has the largest median estimated power, followed by the DESeq2, SAMSeq and NOISeq methods. The NOISeq method also shows larger variability in estimated power than the other methods. The total number of discoveries follows the same pattern as the estimated power. EBSeq has the largest median total rejections, followed by SAMSeq, DESeq2, and NOISeq. The edgeR, DESeq, baySeq, and Voom methods have very little power and reject very few hypotheses when sample size is equal to 3 in each group. The SDs of total discoveries from NOISeq and SAMSeq methods are much larger than the SDs from all other methods, which indicates low stability for NOISeq and SAMSeq methods. A comparison of the eight RNA-seq differential analysis methods in FDR, power, total discoveries, and SD of total discoveries for equal and unequal library sizes, indicates that the differential analyses are unaffected by sequencing depth for all eight methods. However, slight differences are observed between two different approaches in the edgeR method. The edgeR GLM approach has slightly larger FDR estimates than the edgeR Exact approach when library sizes are equal, while the edgeR Exact approach has slightly larger FDR estimates than the edgeR GLM approach when library sizes are unequal.

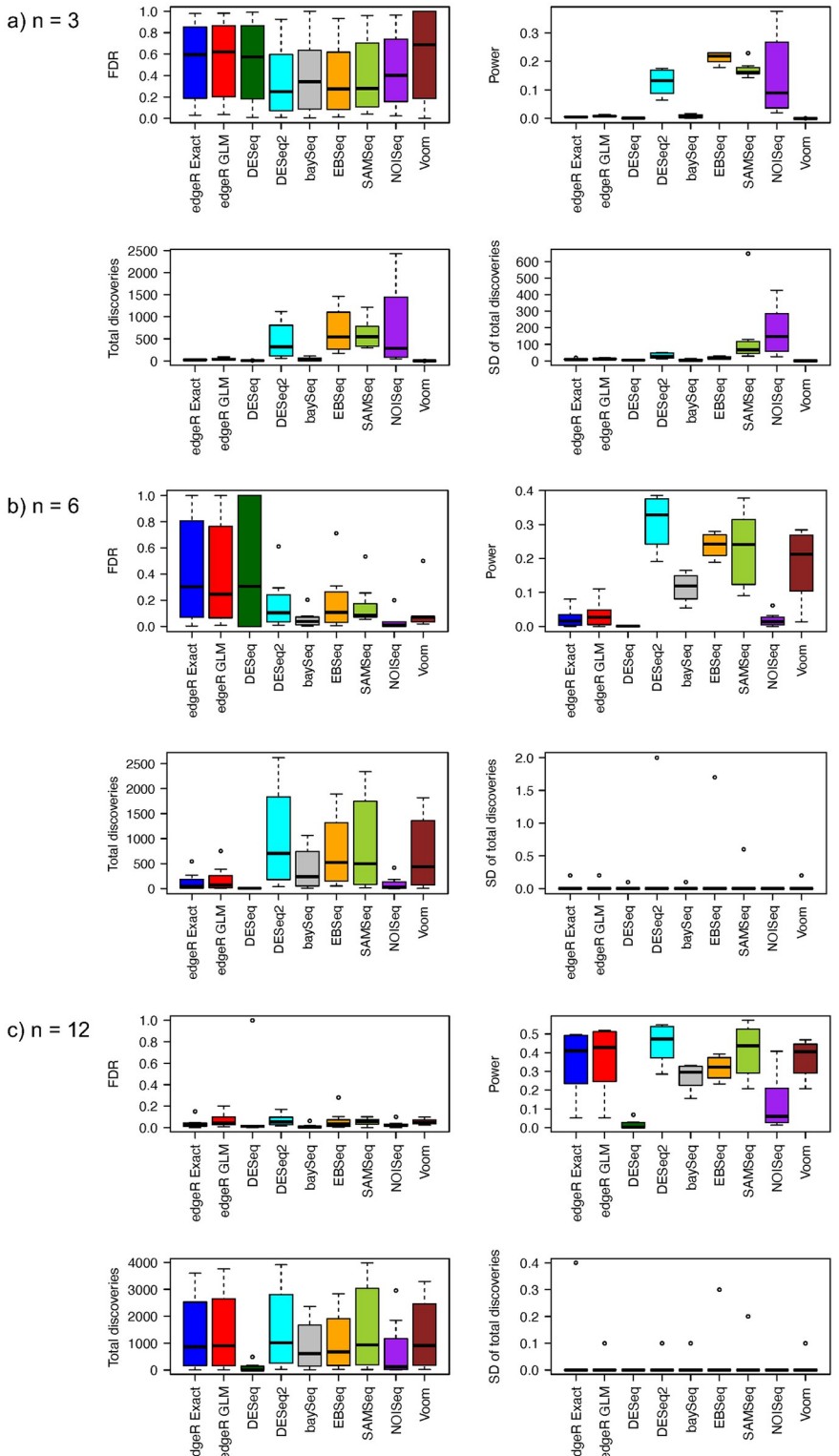

**Fig 1. Estimated FDRs, powers, means of total discoveries, and SD of total discoveries from different RNA-Seq differential analysis methods with negative binomial distribution assumption and equal library size for a)** *n* = 3, **b)** *n* = 6, **and c)** *n* = 12.

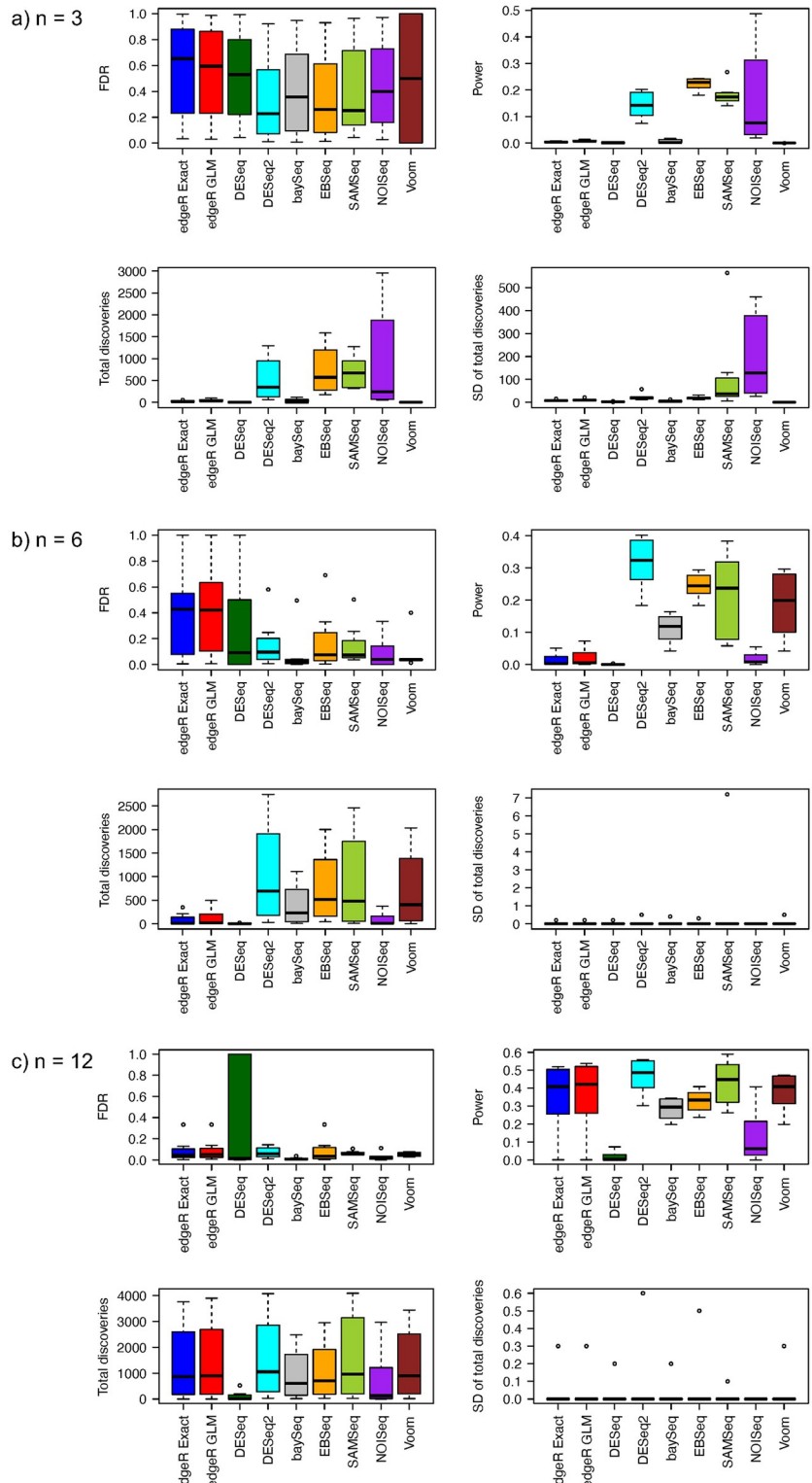

**Fig 2. Estimated FDRs, powers, means of total discoveries, and SD of total discoveries from different RNA-Seq differential analysis methods with negative binomial distribution assumption and unequal library size for a)** *n* = 3, b) *n* = 6, and c) *n* = 12.

When sample size increases to 6 in each group, the comparisons in FDR, power, total discoveries, and SD of total discoveries show different patterns (Figs 1(b) and 2(b)). The edgeR and DeSeq methods have relatively larger FDR estimates than all other methods. We notice the estimates of FDR decrease for all methods when sample size increases from 3 to 6 in each group (S1 Table). The medians of FDR estimates are larger for both edgeR approaches when library sizes are unequal, while the median of FDR estimates is larger for the DESeq method when library sizes are equal. The variability in FDR estimates of the baySeq and NOISeq methods seems larger for unequal library sizes than for equal library sizes. The edgeR, DESeq2, baySeq, SAMSeq, and Voom methods have increased power when sample size increases to 6 (S2 Table). The power of the DESeq and NOISeq methods remains low even when sample size is doubled in each group. The power of the EBSeq method increases only slightly when sample size increases from 3 to 6 in each group. Among all the methods, DESeq2 shows the greatest power, followed by the EBSeq, SAMSeq, Voom, and baySeq methods. The total number of discoveries have similar pattern as the power. The DESeq2 method has the largest median total discoveries, followed by the EBSeq, SAMSeq, Voom, and baySeq methods. The edgeR, DESeq, and NOISeq methods have less median total discoveries. Regarding stability, the SD of total discoveries for all methods ranges from 0 to 2. The stability of the NOISeq and SAMSeq methods improves greatly when the sample size increases to 6 in each group. The range of SD of total discoveries for the SAMSeq method is larger for unequal library sizes than for equal library sizes. The SD of total discoveries for all other methods remains invariant to library sizes.

The performance of all methods shows a different pattern when sample size increases to 12 in each group (Figs 1(c) and 2(c)). The median of estimated FDR for all methods greatly improves as sample size increases from 6 to 12 in each group. No significant differences are observed in FDR control among methods, except the DESeq method with its outlying estimated FDR (S1 Table). The power of all methods increases as the sample size increases. The DESeq2 method still has the largest median power, followed by the SAMSeq, edgeR, Voom, EBSeq, baySeq, NOISeq, and DESeq methods (S2 Table). The DESeq method still shows low power even with large sample size (12 in each group). The highest median power from the DESeq2 method is around 0.5. The number of total discoveries shows a pattern similar to power. The DESeq and NOISeq methods have fewer total discoveries than other methods. The SD of total discoveries for all methods decreases. The maximum SD of total discoveries for all methods decreases from 2 ($n$ = 6 in each group) to 0.4 ($n$ = 12 in each group). Besides the small differences in the maximum SD of total discoveries, no differences are evident in FDR control, power, and total discoveries in the case of equal library sizes or that of unequal library sizes.

**Simulation results from log-normal distribution.** With unequal library sizes, the FDR, power, total discoveries, and SD of total discoveries from simulations assuming log-normal distributed count data are shown in Fig 3. When the sample size is as small as 3 in each group, most RNA-seq differential analysis methods show much better FDR control for log-normal distributed data than for negative binomial distributed data except the Voom method (S3 Table). Besides the Voom method, the NOISeq method shows relatively larger FDR than other methods. The DESeq, DESeq2, EBSeq, and baySeq methods show relatively good FDR control. The power from all methods are much higher for log-normal distributed data than for negative binomial distributed data. Among them, the Voom method has the largest power and total discoveries. The NOISeq, SAMSeq, and DESeq methods have relatively higher power than other methods. The EBSeq method has relatively less power than all other methods. The DESeq2 method has comparable power to the edgeR and baySeq method. All remaining methods have

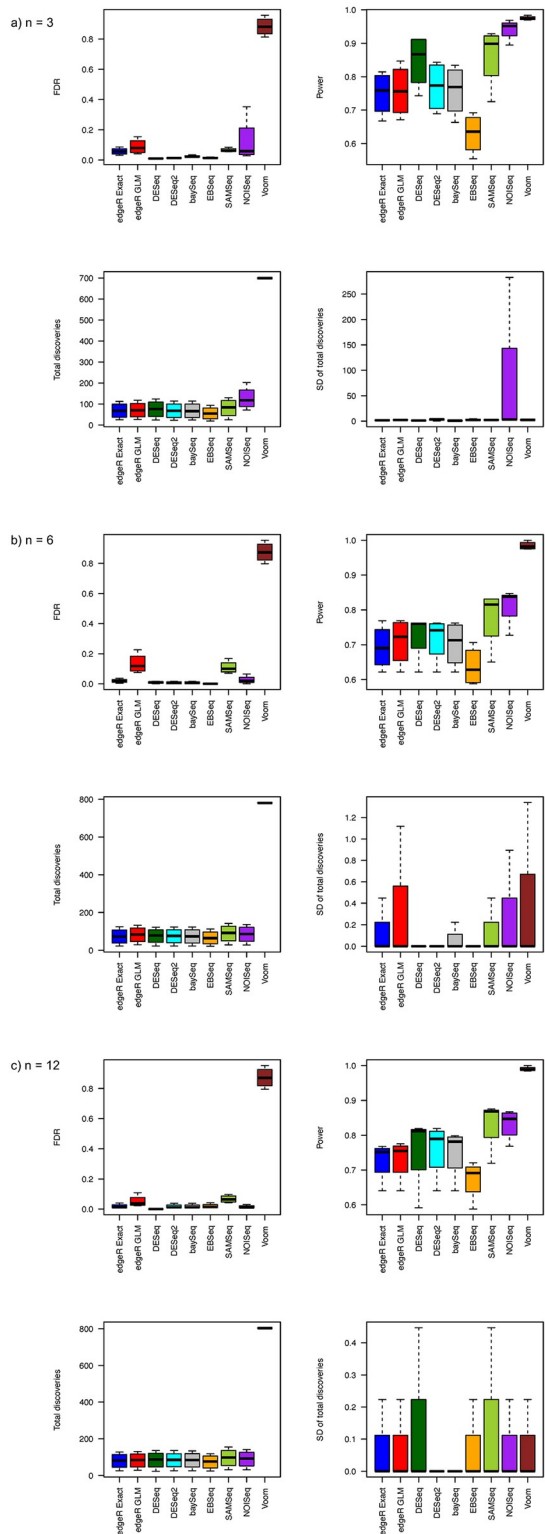

**Fig 3. Estimated FDRs, powers, means of total discoveries, and SD of total discoveries from different RNA-Seq differential analysis methods with log-normal distribution assumption and equal library size for a)** $n = 3$, **b)** $n = 6$, **and c)** $n = 12$.

comparable number of total discoveries. For stability, the NOISeq method shows lower stability than all other method with the largest SD of total discoveries.

The FDR, power, and stability pattern changes when sample size increase to 6 in each group. Voom still has the largest FDR among all compared methods. The edgeR GLM, SAM-Seq, and NOISeq methods also show relatively larger FDR than all other methods (S3 Table). The edgeR Exact, DESeq, DESeq2, baySeq, and EBSeq methods control the FDR within 5%. The Voom method still have the largest power followed by the NOISeq and SAMSeq method. The power of the edgeR, DESeq, DESeq2, and baySeq methods are comparable and the EBSeq method shows the lowest power among all methods. Except the Voom method, all other methods show similar number of total discoveries. For stability, the DESeq, DESeq2, and EBSeq methods show relatively better stability than all other methods.

When sample size further increases to 12 in each group, the pattern of FDR, power, and stability is slightly changed from when the sample size is 6 in each group. The Voom method has the largest FDR, followed by the SAMSeq and edgeR GLM methods (S3 Table). All other methods have relatively small FDR. The Voom method still have the largest power, followed by the SAMSeq and NOISeq method. The EBSeq method has the lowest power among all methods. The edgeR, DESeq, DESeq2, and baySeq methods have comparable power. For total discoveries, the Voom method still have the largest number and all other methods have similar numbers. Regarding the stability, the DESeq2 and baySeq methods show relatively better stability than all other methods.

## Real data example

To further examine the apparent test power (the total number of discoveries) and the overlap of discoveries among popular RNA-seq differential analysis methods, we applied all methods to actual summarized RNA-seq count data obtained from the ReCount website [21]. The RNA-seq count data are from the two most commonly-used inbred mouse strains in neuroscience research—C57BL/6J(B6) and DBA/2J(D2) [22]. Measurements on 36536 genes from 10 B6 mouse samples and 11 D2 mouse samples were obtained from the RNA-seq experiment. We first filtered out genes with summed counts across all samples less than 10. After filtering, 11870 genes remain for analysis. We used the TMM method to normalize summarized count data for all genes across all samples. Normalized count data are used as input for all RNA-seq differential analysis methods to identify differentially expressed genes between two mouse strains. The raw *p*-values obtained from all methods are adjusted using the Benjamini-Hochberg procedure [19] to control the FDR at 5%.

The apparent test power results show that the SAMSeq method has the largest number of total discoveries, followed by DESeq2, edgeR GLM, edgeR Exact, Voom, DESeq, baySeq, NOISeq, and finally EBSeq (Fig 4a). The apparent test power results are consistent with our simulation results when the sample size is 12 in each group.

The Venn diagram of significantly differentially expressed genes from each RNA-seq analysis method shows an overlap of 374 genes among the SAMSeq, DESeq2, Voom, edgeR Exact, and EBSeq methods (Fig 5a). The SAMSeq method has the largest number of significant genes that are not identified by other methods. After removing the SAMSeq method from the Venn diagram, the overlap remains at 374 (Fig 5b). The Venn diagrams from the SAMSeq, DESeq2, DESeq, baySeq, and NOISeq methods show an overlap of 255 genes among those methods (Fig 5e). The overlap still remains at 255 after removing the SAMSeq method from the Venn diagram (Fig 5f).

We randomly selected 3 mouse samples from the B6 mouse strain and 3 samples from the D2 mouse strain to examine apparent test power and overlap of all methods for small sample

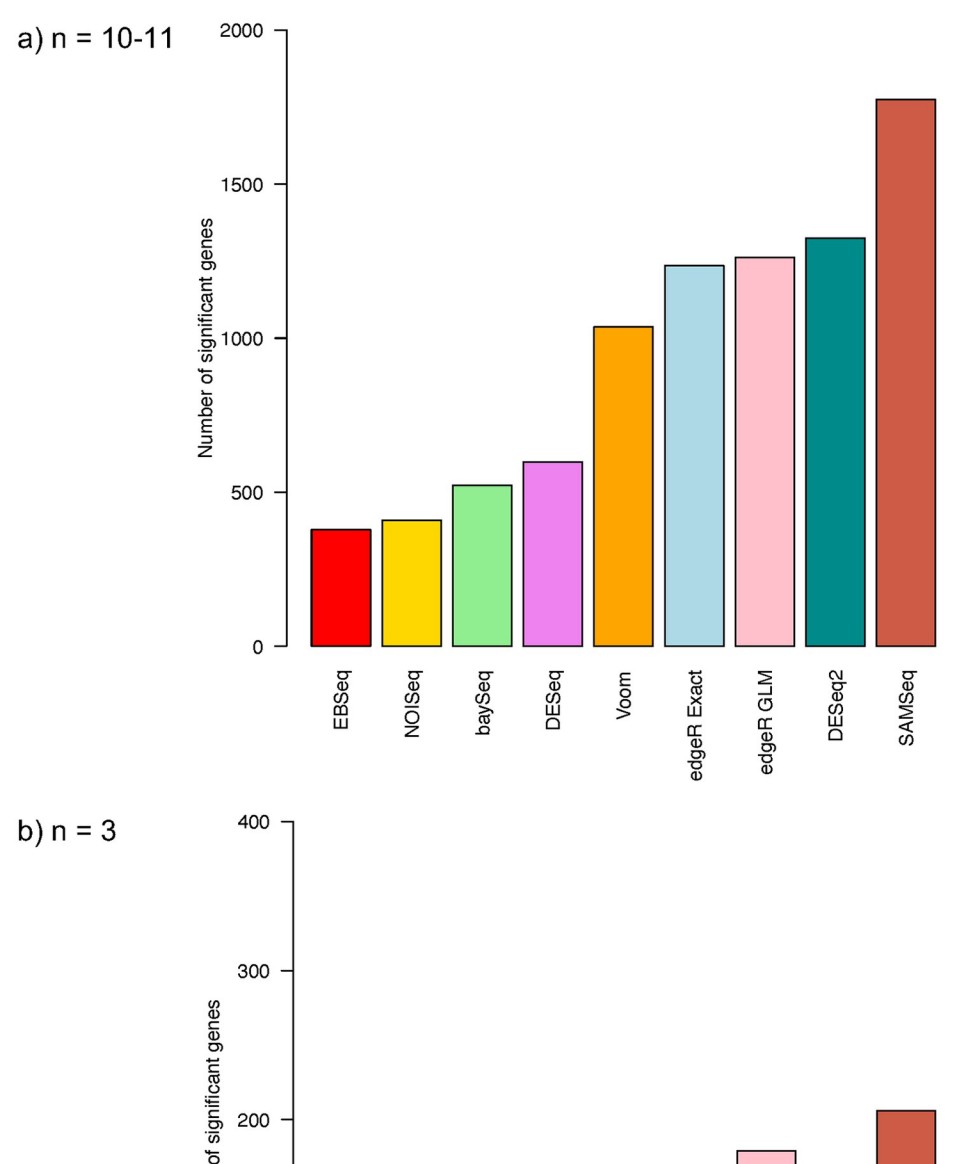

**Fig 4. Number of identified significant genes from bottomly et al.** a) data with $n = 10–11$ samples in each group and b) randomly selected $n = 3$ samples in each group.

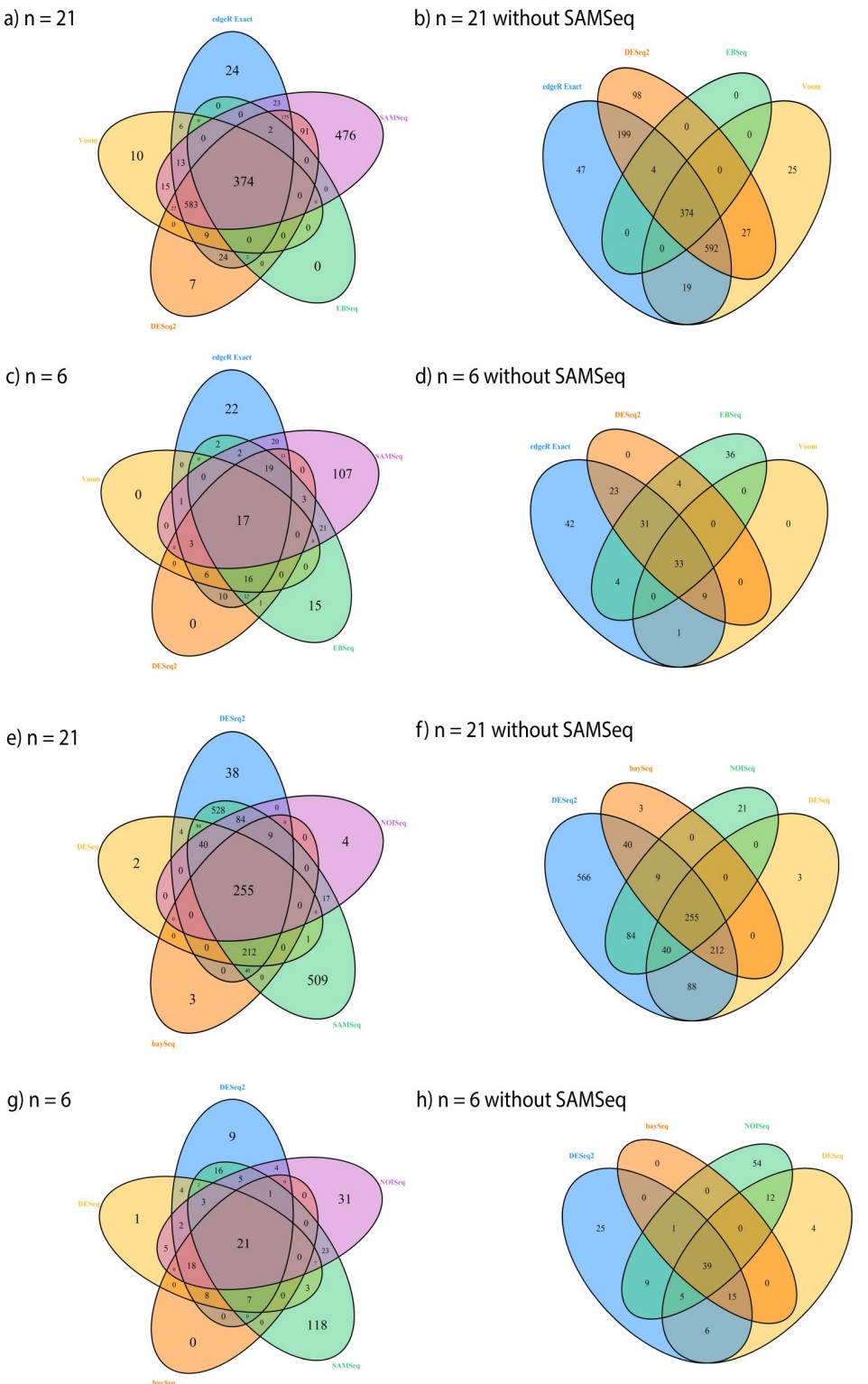

**Fig 5. Venn diagram of selected significant genes from different RNA-seq differential analysis methods with 21 samples and randomly selected 6 samples.** (a)-(d) use significant genes from the edgeR Exact, SAMSeq, EBSeq, DESeq2, and Voom methods; (e)—(h) use significant genes from the DESeq2, NOISeq, SAMSeq, baySeq, DESeq methods.

size data. Using the same filtering criteria, 10815 genes remained for differential analysis with FDR controlled at 5%. After being normalized with the TMM method, count data for each gene were analyzed using each method. With sample size at 3 in each group, the SAMSeq method shows the largest number of total discoveries, followed by the edgeR GLM, edgeR Exact, NOISeq, EBSeq, DESeq2, DESeq, baySeq, and Voom methods (Fig 4b). There is an overlapped of only 17 genes among the SAMSeq, DESeq2, Voom, edgeR Exact, and EBSeq methods (Fig 5c). The SAMSeq method has the largest number of significant genes that are not found by other methods. The overlap increases to 33 genes when the SAMSeq method is removed from the Venn diagram (Fig 5d). Twenty-one of the identified genes overlaps among the SAMSeq, DESeq2, DESeq, baySeq, and NOISeq methods (Fig 5g). After removing the SAMSeq method, the overlap increases from 21 to 39 genes (Fig 5h). The two edgeR methods identified similar genes. Of 143 genes identified by the edgeR Exact method and 179 genes identified by the edgeR GLM method, with sample size of 3 in each group, 143 genes were identified by both methods.

## Discussion

We evaluated eight commonly used RNA-seq differential analysis methods in this study through both simulation studies and real RNA-seq data examples. We compared the FDR control, power, apparent test power, and stability of eight methods under different scenarios with varied library sizes, distribution assumptions and sample sizes. Our studies show the library size does not have much effect on performance, which is due to the adjustment of library size in all methods. Previous comparisons on RNA-seq differential analysis methods are either based on negative binomial distributed data or real RNA-seq data [2, 16, 25, 26]. To eliminate potential bias from distribution assumptions in our simulation studies, we simulated our summarized RNA-seq count data from both negative binomial distribution and log-normal distribution. The simulation results show different performances of all methods under these two different distribution assumptions. Meanwhile, sample sizes also have significant effects on the performance of all methods.

The FDR is much better controlled for all methods except the Voom method, when the count data follows the log-normal distribution rather than the negative binomial distribution especially for data with small sample size (3 in each group). The power of all methods are much higher for log-normal distributed than for negative binomial distributed count data. When sample size is 3 in each group, the Voom method shows relatively higher power but much worse FDR control than all other methods for log-normal distributed data. In contrast, the power for the Voom method is close to zero and its FDR control is relatively worse than all other methods for negative binomial distributed data. When the sample size increases to 6 or 12 in each group, the performance of Voom is improved for negative binomial distributed data, while remains the same for log-normal distributed data. All other methods have relatively similar performance for either negative binomial distributed data or log-normal distributed data.

When the sample size equals to 3 in each group, the EBseq method performs best for negative binomial distributed data and the DESeq method performs best for log-normal distributed data, considering FDR control, power, and stability. The DESeq2 method also performs well with relatively better FDR control and higher power than most of other methods for both negative binomial and log-normal distributed data. When the sample size is small, the SAMSeq and NOISeq methods have low stability for negative binomial distributed data, while the NOISeq method still have low stability for log-normal distributed data, which might be a consequence of their nonparametric approaches. We notice the estimated large FDR and low

power for all methods when sample size equals 3 for negative binomial distributed data, thus caution is needed when interpreting analysis results as noted by Soneson and Delorenzi [16] in their comparison of RNA-seq differential analysis methods.

When the sample size equals to 6 in each group, the DESeq2 method shows the best performance for negative binomial distributed data, while both the DESeq and DESeq2 methods show the best performance for log-normal distributed data, considering FDR control, power, and stability. The EBSeq, SAMSeq, Voom, and baySeq methods also perform well in case for negative binomial distributed data. Previous simulations showed the SAMSeq method and methods with variance stabilization transformations perform well in various scenarios [16]. The DESeq2, EBSeq, Voom, and baySeq methods all adapt the empirical Bayes approach to stabilize the variance estimates of the RNA-seq count data. The SAMSeq method takes a resampling approach and uses the geometric mean to estimate variance, which is less than or equal to the arithmetic mean (the arithmetic mean is also the variance in Poisson distribution), thus the estimated variance is reduced by using the geometric mean. We also identify the liberty control of FDR by the edgeR and DESeq method when the sample size is 6 in each group, just as found in previous comparisons in RNA-seq analysis methods [2, 25, 26]. For log-normal distributed data, all methods perform relatively well except the poor FDR control from the Voom method.

When sample size is large (12 in each group), all methods demonstrate improved performance, with the exception of the DESeq method that continues to show low power for negative binomial distributed data. This might be due to the exact approach the DESeq method takes, producing a conservative result and low power even with large samples. We notice a large improvement in FDR control and stability when the sample size increases from 3 to 6 in each group. FDR control and stability are further improved when we increase sample size from 6 to 12 in each group. Sample size is always a very important factor to be considered in study design of RNA-seq experiments. With the cost of sequencing reducing in each year, we advise consideration of larger sample size with at least 6 samples in each group in RNA-seq experiment to minimize false discoveries and increase power. For log-normal distributed data, all methods perform similar to that when sample size is 6 in each group.

The real RNA-seq data examples showed apparent test power consistent with our simulation studies. The Venn diagrams of discoveries from different RNA-seq analysis methods highlight the need for better RNA-seq differential analysis methods and a combined use of different RNA-seq analysis methods. The small overlaps among different RNA-seq differential analysis methods, especially when the sample size is small, calls for caution when interpreting the validity of identified genes. The SAMseq method overlaps least with other methods, which might due to its resampling approach for selecting differentially expressed genes. A recent study showed that the intersections between differentially expressed genes detected by Cuffdiff2, edgeR, DESeq2 and Two-stage Poisson Model (TSPM) is zero, after controlling the FDR at 5% using the Benjamini-Hochberg method [27]. In a situation where many genes have been identified as differentially expressed, it might be a good exercise to select top candidates identified by several different RNA-seq differential analysis methods for further validation. The power of RNA-seq differential analysis methods still remains at around 50% even when sample size is equal to 12 in each group, which indicates the need for new RNA-seq differential analysis methods to further increase the power and improve the validity of the identified significant genes. We noticed that the number of differentially expressed genes identified from the real data have large variations among the negative-binomial based models. This large variation might be due to the different ways of estimating the standard deviations in the models, which lead to the power differences in different models. We also observed that edgeR ranks amongst the lowest for power in the simulations at n = 6 and n = 3 sample per group, while edgeR ranks

highly for number of significant genes identified in the real data. This difference might be due to the distribution differences between the simulated data and real data. Our simulated data assumes independence between genes, which might not be the situation in the real data.

A limitation of our simulation study is the lack of consideration of structural correlation among genes as our random sampling from negative-binomial and log-normal distributions assumes independence between samples, which is not the real situation in gene expression. Future simulations considering the gene correlations are needed to further evaluate the performance of RNA-seq differential analyses methods. Another limitation is that we only compared the gene differential analysis methods that fall in our interest. Our current methods comparison might miss some new methods (such as the Sleuth method) that are getting popular in gene differential analyses. Future updated comparisons of RNA-seq differential analysis methods are needed to incorporate newly developed methods that are getting popular in recent years.

## Conclusion

We evaluate eight commonly used RNA-seq differential analysis methods in R/Bioconductor including the edgeR, DESeq, DESeq2, baySeq, EBSeq, NOISeq, SAMSeq, and Voom methods. We include two approaches in the edgeR method: edgeR Exact and edgeR GLM. Our simulation results show that the EBSeq method has the best performance for negative binomial distributed data in terms of FDR control, power, and stability, when the sample size is small (3 in each group). The DESeq2 method performs the best when the sample size is 6 or higher in each group for negative binomial distributed data. For log-normal distributed data, both the DESeq and DESeq2 methods perform relatively better than other methods in terms of FDR control, power, and stability across all sample sizes. The small number of identified differentially expressed genes overlapped among different RNA-seq differential analysis methods indicates the great need for new RNA-seq differential analysis methods.

## Supporting information

**S1 Table. Estimated FDR of compared RNA-seq differential analysis methods from negative binomial distributed RNA-seq count data.**
(PDF)

**S2 Table. Estimated power of compared RNA-seq differential analysis methods from negative binomial distributed RNA-seq count data.**
(PDF)

**S3 Table. Estimated FDR and power of compared RNA-seq differential analysis methods from log-normal distributed RNA-seq count data.**
(PDF)

## Acknowledgments

We thank the Center for Integrated Research Computing at the University of Rochester for providing high performance computing resources. We also would like to thank the anonymous reviewers for their insightful comments and suggestions that helped to further improve our manuscript.

## Author Contributions

**Conceptualization:** Dongmei Li, Zidian Xie.

**Data curation:** Dongmei Li.

**Formal analysis:** Dongmei Li.

**Funding acquisition:** Dongmei Li.

**Investigation:** Dongmei Li.

**Methodology:** Dongmei Li.

**Project administration:** Dongmei Li.

**Resources:** Dongmei Li.

**Software:** Dongmei Li.

**Validation:** Dongmei Li.

**Visualization:** Dongmei Li.

**Writing – original draft:** Dongmei Li, Martin S. Zand, Timothy D. Dye, Maciej L. Goniewicz, Irfan Rahman, Zidian Xie.

**Writing – review & editing:** Dongmei Li, Martin S. Zand, Timothy D. Dye, Maciej L. Goniewicz, Irfan Rahman, Zidian Xie.

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
