## [Decision Letter · Decision Letter 0]

18 Jul 2022

PONE-D-22-03667An evaluation of RNA-seq differential analysis methodsPLOS ONE

Dear Dr. Li,

Thank you for submitting your manuscript to PLOS ONE. After careful consideration, we feel that it has merit but does not fully meet PLOS ONE’s publication criteria as it currently stands. Therefore, we invite you to submit a revised version of the manuscript that addresses the points raised during the review process. As you can see, the reviewers are broadly supportive, and raise minor points for you to address. Please do address these in your revised manuscript.

We look forward to receiving your revised manuscript.

Kind regards,

Dov Joseph Stekel

Academic Editor

PLOS ONE

“This work is supported by the University of Rochester's Clinical and Translational Science Award (CTSA) number UL1 TR000042, UL1 TR002001, and U24TR002260 from the National Center for Advancing Translational Sciences of the National Institutes of Health (Drs. Li and Zand). Dr. Zand is also supported by the National Institute of Allergy and Infectious Diseases and the National Institute of Immunology, grant numbers AI098112 and AI069351. This study was supported by the National Institute of Environmental Health Sciences with grant number NIH 1R21ES032159-01A1 and National Institute on Aging with grant number NIH 1U54AG075931-01 (Drs. Li and Rahman). This study was supported by the grants from the WNY Center for Research on Flavored Tobacco Products (CRoFT) under cooperative agreement U54CA228110 which is supported by the National Cancer Institute of the National Institutes of Health (NIH) and the Food and Drug Administration (FDA) Center for Tobacco Products (Drs. Li, Goniewicz, Rahman, Xie). The content is solely the responsibility of the authors and does not necessarily represent the official views of the NIH and FDA.”

“This work is supported by the University of Rochester’s Clinical and Translational 496 Science Award (CTSA) number UL1 TR000042, UL1 TR002001, and U24TR002260 497 from the National Center for Advancing Translational Sciences of the National Institutes 498 of Health (Drs. Li and Zand). Dr. Zand is also supported by the National Institute of 499 Allergy and Infectious Diseases and the National Institute of Immunology, grant 500 numbers AI098112 and AI069351. This study was supported by the National Institute 501 of Environmental Health Sciences with grant number NIH 1R21ES032159-01A1 and 502 National Institute on Aging with grant number NIH 1U54AG075931-01 (Drs. Li and 503 Rahman). This study was supported by the grants from the WNY Center for Research 504 on Flavored Tobacco Products (CRoFT) under cooperative agreement U54CA228110 505 which is supported by the National Cancer Institute of the National Institutes of Health 506 (NIH) and the Food and Drug Administration (FDA) Center for Tobacco Products (Drs. 507 Li, Goniewicz, Rahman, Xie). The content is solely the responsibility of the authors and 508 does not necessarily represent the official views of the NIH and FDA.”

“This work is supported by the University of Rochester's Clinical and Translational Science Award (CTSA) number UL1 TR000042, UL1 TR002001, and U24TR002260 from the National Center for Advancing Translational Sciences of the National Institutes of Health (Drs. Li and Zand). Dr. Zand is also supported by the National Institute of Allergy and Infectious Diseases and the National Institute of Immunology, grant numbers AI098112 and AI069351. This study was supported by the National Institute of Environmental Health Sciences with grant number NIH 1R21ES032159-01A1 and National Institute on Aging with grant number NIH 1U54AG075931-01 (Drs. Li and Rahman). This study was supported by the grants from the WNY Center for Research on Flavored Tobacco Products (CRoFT) under cooperative agreement U54CA228110 which is supported by the National Cancer Institute of the National Institutes of Health (NIH) and the Food and Drug Administration (FDA) Center for Tobacco Products (Drs. Li, Goniewicz, Rahman, Xie). The content is solely the responsibility of the authors and does not necessarily represent the official views of the NIH and FDA.”

“The authors declare that they have no competing interests.”

Reviewers' comments:

Reviewer's Responses to Questions

**Comments to the Author**

1. Is the manuscript technically sound, and do the data support the conclusions?

Reviewer #1: Yes

Reviewer #2: Yes

2. Has the statistical analysis been performed appropriately and rigorously? 

Reviewer #1: Yes

Reviewer #2: Yes

3. Have the authors made all data underlying the findings in their manuscript fully available?

Reviewer #1: Yes

Reviewer #2: No

4. Is the manuscript presented in an intelligible fashion and written in standard English?

Reviewer #1: Yes

Reviewer #2: Yes

5. Review Comments to the Author

Reviewer #1: Li et al have produced a robust and thorough review of differential expression analysis tools. The article is very well written, included all technical descriptions of the tools and methods they employ and gives a well reasoned discussion and conclusion for each.

I have a few very minor things which I wold like to see addressed:

On line 42 you describe DeSeq2 as recent, I would disagree as it was published in 2014. I think this needs rewording.

I think in the introduction more emphasis could be made on previous published comparisons being 8-10 years old and it being time for an update.

I think a layman's description of Power and Stability would be of benefit to newer bioinformaticians and would widen the scope of the readership.

Figure 3 axis titles are a bit squashed compared to others, would be nice to see them consistent (however this may be an artefact of the submission).

Figure 4 legend needs to include A and B descriptions o match the figure.

Finally why has Sleuth been committed? Rationale needs adding to discussion on choice of tools. As "popular" depends on the audience.

Reviewer #2: The authors have conducted an interesting study comparing popular tools for gene differential expression analysis. Using a combination of simulations and RNA-seq data from mice they compare the power, FDR and stability of results for each tool.

The results are clearly presented and address relevant questions for RNA-seq analysts.

The discussion would benefit from some comments on why the number of DE genes vary so much among the negative-binomial models in the mouse data. Why does EdgeR rank amongst the lowest for power in the simulations at n=6 and n=3 samples, but ranks highly for number of significant genes identified in the real data?

The discussion also needs to mention the limitations of the simulations used. In particular, random sampling from negative-binomial and log-normal distributions assumes independence between samples, which is unrealistic in the case of gene expression. This approach does not take the structural correlation among genes into account.

It would also be appreciated if the code and simulation data could be made available on a public repository, such as Zenodo or Github. Having it available upon request by email introduces an unnecessary hurdle, especially in light of the recent paper by Gabelica et al. https://doi.org/10.1016/j.jclinepi.2022.05.019

Some minor points to address:

Line 240 — "have very little power and reject very few genes" doesn't make sense here, please clarify. Does this refer to rejecting hypotheses?

Line 341 — "the sample size is 12 in each group" doesn't fit with Figure 4a containing n=21 total. For consistency with Figures 1-3, it would be clearer to stick to the "n=samples per group" format rather than "n=total samples".

6. PLOS authors have the option to publish the peer review history of their article (what does this mean?). If published, this will include your full peer review and any attached files.

Reviewer #1: **Yes: **Adam Mark Blanchard

Reviewer #2: No

---

## [Author Response · Author response to Decision Letter 0]

17 Aug 2022

Overview of responses to reviewers

Manuscript ID: PONE-D-22-03667

An evaluation of RNA-seq differential analysis methods

PLOS ONE

We appreciate the many suggestions and comments from the reviewers. We have revised our manuscript to incorporate reviewers’ comments and suggestions. The details are listed below.

5. Review Comments to the Author

Reviewer #1: Li et al have produced a robust and thorough review of differential expression analysis tools. The article is very well written, included all technical descriptions of the tools and methods they employ and gives a well reasoned discussion and conclusion for each.

I have a few very minor things which I would like to see addressed:

On line 42 you describe DeSeq2 as recent, I would disagree as it was published in 2014. I think this needs rewording.

Response: Thanks for the suggestions. We agree with the reviewer and have reworded the sentence in our revised manuscript.

“DESeq2 is a successor to the DESeq method with flexibility to accommodate more complexed study design of sequencing experiments”.

I think in the introduction more emphasis could be made on previous published comparisons being 8-10 years old and it being time for an update.

Response: Great suggestions. We have emphasized that the previous published comparisons are 8-10 in our revised manuscript.

“As more statistical methods for gene differential analyses have been developed in recent years, there is a need for an update on the differential analysis methods comparisons”.

I think a layman's description of Power and Stability would be of benefit to newer bioinformaticians and would widen the scope of the readership.

Response: Great suggestions. We have added the description of Power and Stability using layman language in our revised manuscript.

“Power is defined as the expected proportion of identified differentially expressed genes among all the truly differentially expressed genes, given at least one gene is truly differentially expressed in the data. Stability is defined as the standard deviation of total rejections.”

Figure 3 axis titles are a bit squashed compared to others, would be nice to see them consistent (however this may be an artefact of the submission).

Response: Thanks for the suggestions. We have revised Figure 3 axis titles to make them consistent with other Figure axis titles.

Figure 4 legend needs to include A and B descriptions to match the figure.

Response: Thanks for the suggestions. A and B descriptions are added.

Finally why has Sleuth been committed? Rationale needs adding to discussion on choice of tools. As "popular" depends on the audience.

Response: Thanks for the suggestions. We have added this into our limitations in the Discussion section as the methods we chose are “popular” methods in our opinion, which might not be considered as “popular” methods by others.

“Another limitation is that we only compared the gene differential analysis methods that fall in our interest. Our current methods comparison might miss some new methods (such as the Sleuth method) that are getting popular in gene differential analyses. Future updated comparisons of RNA-seq differential analysis methods are needed to incorporate newly developed methods that are getting popular in recent years”.

Reviewer #2: The authors have conducted an interesting study comparing popular tools for gene differential expression analysis. Using a combination of simulations and RNA-seq data from mice they compare the power, FDR and stability of results for each tool.

The results are clearly presented and address relevant questions for RNA-seq analysts.

The discussion would benefit from some comments on why the number of DE genes vary so much among the negative-binomial models in the mouse data. Why does EdgeR rank amongst the lowest for power in the simulations at n=6 and n=3 samples, but ranks highly for number of significant genes identified in the real data?

Response: Thanks for the suggestions. We have added comments on why the number of DE genes vary so much among the negative-binomial models in the mouse data as well as why EdgeR rank amongst the lowest for power in the simulations at n = 6 and n = 3 samples but ranks highly for number of significant genes identified in the real data.

“We noticed that the number of differentially expressed genes identified from the real data have large variations among the negative-binomial based models. This large variation might be due to the different ways of estimating the standard deviations in the models, which lead to the power differences in different models. We also observed that edgeR ranks amongst the lowest for power in the simulations at n = 6 and n = 3 sample per group, while edgeR ranks highly for number of significant genes identified in the real data. This difference might be due to the distribution differences between the simulated data and real data. Our simulated data assumes independence between genes, which might not be the situation in the real data.”

The discussion also needs to mention the limitations of the simulations used. In particular, random sampling from negative-binomial and log-normal distributions assumes independence between samples, which is unrealistic in the case of gene expression. This approach does not take the structural correlation among genes into account.

Response: Thanks for the comments. We have added the limitation of our simulation study in the discussion section in our revised manuscript.

“A limitation of our simulation study is the lack of consideration of structural correlation among genes as our random sampling from negative-binomial and log-normal distributions assumes independence between samples, which is not the real situation in gene expression. Future simulations considering the gene correlations are needed to further evaluate the performance of RNA-seq differential analyses methods. ”

It would also be appreciated if the code and simulation data could be made available on a public repository, such as Zenodo or Github. Having it available upon request by email introduces an unnecessary hurdle, especially in light of the recent paper by Gabelica et al. https://doi.org/10.1016/j.jclinepi.2022.05.019

Response: Thanks for the suggestions. We have deposited the simulation data and code into Github. It can be accessed at https://github.com/DongmeiLi2017/RNA-seq-Analysis-Methods-Comparison. 

Some minor points to address:

Line 240 — "have very little power and reject very few genes" doesn't make sense here, please clarify. Does this refer to rejecting hypotheses?

Response: Thanks for the comments. It refers to rejecting hypotheses. We have changed it to “have very little power and reject very few hypotheses.”

Line 341 — "the sample size is 12 in each group" doesn't fit with Figure 4a containing n=21 total. For consistency with Figures 1-3, it would be clearer to stick to the "n=samples per group" format rather than "n=total samples".

Response: Thanks for the comments. We have changed for format to “n = samples per group” in our revised manuscript.

---

## [Editor Report · Decision Letter 1]

30 Aug 2022

An evaluation of RNA-seq differential analysis methods

PONE-D-22-03667R1

Dear Dr. Li,

We’re pleased to inform you that your manuscript has been judged scientifically suitable for publication and will be formally accepted for publication once it meets all outstanding technical requirements.

Kind regards,

Dov Joseph Stekel

Academic Editor

PLOS ONE
---

## [Editor Report · Acceptance letter]

7 Sep 2022

PONE-D-22-03667R1 

An evaluation of RNA-seq differential analysis methods 

Dear Dr. Li:

I'm pleased to inform you that your manuscript has been deemed suitable for publication in PLOS ONE. Congratulations! Your manuscript is now with our production department. 

Kind regards, 

on behalf of

Dr. Dov Joseph Stekel 

Academic Editor

PLOS ONE